# Characteristics of Bacterial Strains with Desirable Flavor Compounds from Korean Traditional Fermented Soybean Paste (*Doenjang*)

**DOI:** 10.3390/molecules26165067

**Published:** 2021-08-21

**Authors:** Sung-Hun Yi, Sang-Pil Hong

**Affiliations:** Principal Researcher, Division of Strategic Food Research, Korea Food Research Institute (KFRI), 245 Nongsaengmyeong-ro, Iseo-myeon, Wanju-gun 55365, Jeollabuk-do, Korea; sunghunyi@kfri.re.kr

**Keywords:** *Bacillus* sp., *Doenjang* (fermented soybean paste), bacteria screening, flavor, metabolites

## Abstract

To identify and analyze the characteristics of the microorganisms involved in the formation of the desirable flavor of *Doenjang*, a total of 179 strains were isolated from ninety-four *Doenjang* collected from six regions in South Korea, and fourteen strains were selected through a sensory evaluation of the aroma of each culture. The enzyme activities of amylase, protease and lipase was shown in the various strains. *Bacillus* sp.-K3, *Bacillus* sp.-K4 and *Bacillus amyloliquefaciens*-J2 showed relatively high protease activity, at 317.1 U, 317.3 U and 319.5 U, respectively. The *Bacillus* sp.-K1 showed the highest lipase activity at 2453.6 U. In the case of amylase, *Bacillus subtilis*-H6 showed the highest activity at 4105.5 U. The results of the PCA showed that *Bacillus subtilis*-H2, *Bacillus subtilis*-H3, and *Bacillus* sp.-K2 were closely related to the production of 3-hydroxy-2-butanone (23.51%~43.37%), and that *Bacillus subtilis*-H5 and *Bacillus amyloliquefaciens*-J2 were significantly associated with the production of phenethyl alcohol (0.39% and 0.37%). The production of peptides was observed to vary among the *Bacillus* cultures such as Val-Val-Pro-Pro-Phe-Leu and Pro-Ala-Glu-Val-Leu-Asp-Ile. These peptides are precursors of related volatile flavor compounds created in *Doenjang* via the enzymatic or non-enzymatic route; it is expected that these strains could be used to enhance the flavor of *Doenjang*.

## 1. Introduction

Globalization and cultural diversity have greatly increased interest in ethnic foods, motivating food companies to expand into the global food market. Korean *Doenjang* is a traditional fermented soybean paste with a special flavor and healthy effects, as well as a representative daily food deeply embedded in the food culture of Korea. Additionally, it is expected to become a promising food in a world market characterized by the quest for new flavors and beneficial properties [1]. Although *Doenjang* is one of the most important side dishes in Korean cuisine, studies on microbial communities have been limited due to the complexities of the fermentation process.

In general, as shown in Figure 1 below, traditional *Doenjang* is made from *Meju*, which is prepared by soaking, steaming and crushing soybeans, forming them into blocks, and then hanging them up by rice straws to dry for one to three months. After that, the fermented *Meju* is placed in brine and ripened for more than two to three months. After fermentation, the solid part is separated from the liquid, crushed, and then further ripened in a ceramic pot for at least two more months [2].

The primary microorganism involved in the fermentation of *Doenjang* is known as *Bacillus subtilis*. As *Meju*, the main ingredient of *Doenjang*, is fermented in a natural environment in Korea, genus of fungi such as *Rhizopus*, *Mucor,* and *Aspergillus* are also involved in the fermentation process of *Meju* [3,4,5,6].

*Doenjang* has a variety of sensory flavors, such as sweetness, umami, and savory, due to the presence of various amino acids, free sugars, organic acids, and nucleic acids [7,8,9,10,11]. In addition, many studies have documented anticancer, anti-mutagenic, antihypertensive, fibrinolytic, immune, and anti-oxidative activities of *Doenjang* [12,13,14,15,16].

The compounds related to the flavor and functionality of *Doenjang* are known to be affected by microorganisms and fermentation conditions. Although traditional *Doenjang* has better flavor and functionality than commercial products, i.e., modified types of traditional *Doenjang*, further research aimed at improving *Doenjang*’s pungent odor and occasional bitterness is required.

Recently, it has become important to improve the flavor of ethnic foods in view of the global demand for new flavors and health benefits. A number of studies have reported the sensory characteristics [17,18,19], volatiles [18,20,21,22] or non-volatiles [2,7,9,23] of *Doenjang*. In addition, -omics research techniques have been applied to studies on the genomes of microorganisms involved in fermentation, and the correlation between strains and fermented products has been studied. It has been recognized that the flavor-related compounds of fermented soy products are produced by enzymatic or non-enzymatic reactions and by complicated steps depending on the types of strains and the fermentation conditions [21].

On the other hand, studies of long-term fermented *Doenjang* have reported an increase in the content of desirable flavor compounds such as ester like ethyl acetate and lactic acid as well as an increase in functional gamma-aminobutyric acid (*GABA*) [21,24,25,26,27].

Additionally, *Doenjang* aged for three years shows the highest sensory preference [26], so it would be desirable to find a strain capable of producing the preferred flavor compounds from fermented soybean products aged over different periods of time.

We first isolated microorganisms that produce preferable flavor from the collected traditionally made *Doenjang*, and analyzed the enzyme activity, aroma and non-volatile compounds of these cultures of microorganisms to obtain basic data for the production of excellent flavored traditional soybean products using these strains.

## 2. Results

### 2.1. Screening of Strains

As shown in Figure 2, 179 microorganisms isolated from *Doenjang* were identified through 16S rRNA sequencing. Various selective media were used for the microbial screening, and 75% (134 out of 179 species) of the microbes isolated using tryptic soy broth (TSB) were mostly *Bacillus*. *Bacillus* is a microorganism mainly found in traditional Korean fermented foods made with soybeans or soybean paste. Among them, *Bacillus subtilis* is the dominant species, accounting for 38% of *Bacillus*. *Bacillus subtilis* is a microorganism that is easily found, even in old dry grass, and its origin can be deduced from straw used in the process of making *Meju*, the main ingredient of soybean paste.

Based on the 16S rRNA sequencing of the isolates used in this study, fifty-one, twenty-three, thirteen, fourteen, ten, and ten isolates belonging to the species of *Bacillus subtilis*, *Bacillus* sp., *Bacillus amyloliquefaciens*, *Bacillus licheniformis*, *Sporolactobacillus nakayamae, Sporolactobacillus terrae* have been identified, respectively. In the case of yeast, only four isolates belonging to the species of *Candida haemulonis* have been identified.

### 2.2. Enzyme Activity of Microorganisms from Doenjang

Among the strains isolated from *Doenjang* as shown in Table 1, fourteen kinds of microorganisms produced a good aroma. *Bacillus* sp.-K3, *Bacillus* sp.-K4 and *Bacillus amyloliquefaciens*-J2 showed relatively high protease activity in this study, at 317.1 U, 317.3 U and 319.5 U, respectively. As for lipase activity, the *Bacillus* sp.-K1 strain showed the highest value with 2453.6 U, while *Bacillus* sp.-K2, *Bacillus* sp.-K4 and *Bacillus* sp.-K3 showed values of 1639.9 U, 1588.9 U and 1558.7 U, respectively. In the case of amylase activity, *Bacillus subtilis*-H6 strain showed the highest activity at 4105.5 U.

From the above results, *Bacillus subtilis*-H6 showed higher amylase and lipase activity, whereas *Bacillus amyloliquefacience*-J2 showed relatively high protease and lipase activity, and the *Bacillus* sp.-K3 strain showed higher protease, amylase and lipase activities, compared to the other strains. Therefore, it is suggested that strains with high enzyme activity could be closely related to the production of various aromatic compounds during the metabolic process of fermentation.

### 2.3. Descriptive and Volatile Analysis of Cultures

Among the hundreds of *Bacillus* spp. and *Brevibacterium* cultured in TSB, *Bacillus sonorensis*, six species of *Bacillus subtilis*, two species of *Bacillus amyloliquefaciens*, and four species of *Bacillus* sp. and *Brevibacterium frigoritolerans* were selected on the basis of an evaluation of aroma (Table 2).

A descriptive analysis of the strain culture was used to derive sensory characteristics such as butter, cheese, flower, savory aroma, pineapple, bean, peanut, caramel, sweet aroma and wine. Table 2 lists the volatiles identified in the strain culture, and shows their relative peak areas and retention time (RT) in the gas chromatography-mass spectrometry (GC/MS) column. A total of sixteen volatile compounds were identified by SPME, including five alcohols, two aldehydes, three ketones, three acids and three pyrazines. The amount of 3-methyl butanal, acetone, benzaldehyde, acetic acid and 2-methyl propanol was relatively higher in the control than in the strain culture, which shows the main volatile composition of the media. However, these compounds decreased more or less than control (TSB media), and a relatively large amount of 3-hydroxy-2-butanone (acetoin) with butter or cheese notes, 2, 5-dimethyl pyrazine with nutty notes, phenethyl alcohol with rose notes, and 1-butanol were found in most of the strain cultures.

To ascertain a more detailed correlation between strains and volatile compounds produced, the above results (Table 2) were subjected to a principal component (PC) analysis, as shown in Figure 3. The F1 (45.32%) was clearly distinguished from the control (TSB medium, on the positive F1 dimension) and treatment groups (strains culture, on the negative F1 and F2 dimension). In particular, 3-Hydroxy-2-butanone (acetoin), 2-methyl propanoic acid, 1-butanol and ethyl alcohol are major volatiles contributing to the F1 axis. Among strains, the No. 3 (*B. subtilis*-H2), 4(*B. subtilis*-H3), 11(*B.* sp-K2) strains were closely associated with the production of 3-hydroxy-2-butanone (acetoin), 2-methyl propanoic acid, 1-butanol and ethyl alcohol, while the production of phenethyl alcohol and methyl pyrazine showed a closer relation with the No. 6 (*B. subtilis*-H5)*,* 8(*B. amyloliquefacience*-J1), 9(*B. amyloliquefacience*-J2) strains. Additionally, the PC analysis showed that the control group had a significantly close relationship with the volatile compounds such as 3-methyl butanal, acetone, benzaldehyde, acetic acid and 2-methyl propanol.

Though the number of volatiles identified from the above strain culture is fewer than those found in *Doenjang* product, a relatively high concentration of desirable volatile compounds applicable to *Doenjang* such as 3-hydroxy-2-butanone (butter or cheese notes), 2, 5-dimethyl pyrazine (nutty notes) and phenethyl alcohol (rose notes) was detected in most of the selected strain cultures used in this study [21,24].

### 2.4. Non-Volatile Analysis of Culture

Table 3 lists the major metabolites that contribute to separation among the sample groups identified in the cultures, based on their liquid chromatography-mass spectrometry (LC/MS) chromatograms and retention time (RT) shown in the LC/MS columns. Figure 4 shows a heat map for comparison of each compound produced among the strain groups.

A total of twenty metabolites, including ten unknown peaks, were obtained, and there were differences among the six kinds of peptides found among the strains. *Bacillus subtilis*-H2 and *Bacillus subtilis*-H6 generated more peptides, such as Gly-Gly-Val, Val-Pro-Pro-Lys-Ala-Ile, and Val-Ala-Pro-Phe-Pro-Glu, than the other strains. In particular, *Bacillus sonorensis*, *Bacillus sp*.-K4 *and Bacillus sp*.-K1 produced more Val-Val-Pro-Pro-Phe-Leu, whereas *Bacillus sp*.-K3 *and*
*Brevibacterium frigoritolerans* produced more Pro-Ala-Glu-Val-Leu-Asp-Ile. These results are highly consistent with the higher enzyme activity of the screened strains described above (Table 1). Therefore, the above strains could be applied to the development of *Doenjang* with a desirable flavor, and could also provide valuable information for studies of the fermentation process involved in the production of traditional fermented soybean paste (*Doenjang*). Studies on fermentation using these strains are currently under way.

## 3. Discussion

*Bacillus* sp. exists evenly on the surface and inside of *Meju*, and is known to produce various enzymes, such as protease and amylase, during the fermentation process, giving traditional *Doenjang* its unique taste and aroma [28].

*Zygomycetes* such as *Aspergillus oryzae*, *Mucor*, and *Lichtheimia*, *Penicillium*, *Eurotium*, and *Scopulariopsis* are commonly found in *Meju*. Among these, *Aspergillus oryzae* has high enzymatic activity, such as protease and amylase, and is known to play an important role in breaking down proteins and carbohydrates into amino acids and monosaccharides [29].

The lactic acid bacteria (LAB) found in *Meju* produce lactic acid, CO_2_ and alcohols. Yeast is involved in making esters, one of the organoleptic quality factors, via esterification reaction of alcohols and organic acids or fatty acids.

Kim et al. [5] reported that the distribution of microorganisms in *Doenjang* analyzed through pyrosequencing showed that *Bacillus* accounted for 86% of the total microorganisms. However, *Bacillus* accounted for only 75% of the total microorganisms in our result, which indicates that microorganisms cannot be isolated through general screening. Modern technology separates microorganisms that grow in harsh natural environments with such condition as extremely high and low temperatures, high pressure, high and low pH, etc., but it is impossible to separate all microorganisms that exist in simple environments such as fermented foods and place them in a culture dish. This is because the conditions in which individual microorganisms can grow cannot be perfectly implemented in a medium, because the distribution of microorganisms identified through Petri dishes is different from the population of microorganisms identified through metagenome analysis.

Long-term aged soybean products (samples) showed a tendency to decrease in terms of number and types of strains when compared with short-term aged ones, while the amounts of amino nitrogen, free amino acid, lactic acid and some ester compounds, including functional compounds such as *GABA*, daidzein and genistein, increased [21,24,25,26,27]. Therefore, in consideration of the differences in compounds between short-term and long-term fermented soybean pastes, it is possible to find strains that produce a desirable flavor from long-term fermented soybeans.

Proteins are broken down into peptides and amino acids by proteases produced by the fungi and bacteria in *Meju*, while starch is broken down by amylase to make sugar. Various flavoring ingredients such as lactic acid and organic acids produced by lactic acid bacteria and yeast are blended to create the unique taste and aroma of doenjang.

Yoo et al. [29] reported that *Bacillus* sp. produces various enzymes such as protease and amylase in fermented soybeans, and imparts the unique flavor and aroma of traditional soybean products. According to Jo et al. [24], protein is hydrolyzed to the peptide by the proteases in *Doenjang*, and an amine is formed through amino acids, followed by aldehydes, and acid and alcohol compounds. They react with reductants through the Maillard reaction, and produces pyrroles through pyrazines. Eventually, aromatic amino acids form phenol, phenethyl alcohol and benzaldehyde.

Many findings concerning the volatile compounds related to *Doenjang* have already been well documented [21,22], and more than 50% of the volatiles identified are composed of ester, such as ethyl acetate and pyrazine compounds. In one recent study, a headspace SPME-GC/MS analysis of twenty-four domestic certified traditional *Doenjang* products identified relatively high concentrations of ethyl alcohol, isoamyl alcohol and phenethyl alcohol, including benzaldehyde, 3-methyl butanal, and ethyl acetate, as well as pyrazine [21].

From the volatile compound profiles of some of the bacillus strain cultures found in this study, significant quantities of 3-hydroxy-2-butanone (acetoin), 2, 5-dimethyl pyrazine and phenethyl alcohol were mainly produced. 3-hydroxy-2-butanone (acetoin) is one of the compounds that give butter its characteristic flavor. This compound (R form) is produced by bacteria such as *Bacillus subtilis* through decarboxylation of the alpha acetolactate [30]. Further, 2, 5-dimethyl pyrazine, which has a nutty odor, can be produced by both the Maillard reaction and fermentation by bacterium, as is the case for *Doenjang* [21]. On the other hand, phenethyl alcohol, known as an aromatic alcohol with a rose-like odor, originates from phenyl alanine. The three aforementioned volatile compounds seem to be closely related to various enzyme activities in strains that can produce precursors of volatiles such as amino acid, fatty acids and free sugars, as described above.

There have been many studies involving an investigation of the volatile compounds responsible for the most important aromatic profiles of soy sauce, based on such indices as threshold, odor activity value (OVA), and the flavor dilution (FD) factor [31,32]. However, the correlation between volatile compounds and the quality of soy sauce remains unclear.

Even though the strains screened in this study did not produce as many volatile compounds as fermented soybean products [22], they are expected to provide a better aroma than common traditional fermented soybean products whose volatile flavor is produced by native organisms, if they are applied to the initial stage or middle stage or both stage of the fermentation process of *Doenjang*.

The production of peptides was observed to vary among the different strain cultures, with *Bacillus subtilis*-H2, and *Bacillus subtilis*-H6 generating more peptides than the other strains. In particular, *Bacillus sonorensis*, *Bacillus* sp.-K4 and *Bacillus* sp.-K1 produced more Val-Val-Pro-Pro-Phe-Leu, while *Bacillus* sp.-K3 and *Brevibacterium frigoritolerans* produced more Pro-Ala-Glu-Val-Leu-Asp-Ile. These peptides (non-volatile compounds) are also precursors of related volatile flavor compounds created in soybean fermented products via the enzymatic or non-enzymatic route [24].

Therefore, the screened strains in this study are thought to be more advantageous for the production of *Doenjang* (fermented soybean paste) or *Ganjang* (soy sauce) with a more, improved pleasant aroma characterized by buttery, nutty or rose like note.

In this study, strains capable of producing relatively high butter, rose, and nutty flavor in traditional *Doenjang* were obtained. Since these flavoring ingredients can have a positive effect on increasing the flavor of food, it is expected that it will be of great help in improving the flavor of traditional soybean products if applied to the current manufacturing of traditional soybean products. In the future, it is expected that additional research on the stable production of these components during fermentation will be required by conducting a test for applying the fermentation with the above strains.

## 4. Materials and Methods

### 4.1. Doenjang Products for Strain Screening

*Doenjang* products were purchased from the domestic market and stored at 4 °C before use. Ninety-four *Doenjang* were used in total, 34 from Gyeonggi province, 5 from Gangwon province, 26 from Chungcheong province, 7 from Gyeongsang province, 15 from Jeolla province, and 7 from Jeju Island, South Korea.

The *Doenjang* used in this study was 45 *Doenjang* aged 1–5 years, 37 *Doenjang* aged 6–10 years, 8 *Doenjang* aged 11–19 years, and 1 *Doenjang* aged more than 30 years.

### 4.2. Media and Screening

A tryptic soy broth was used to isolate *Bacillus*, a De Man, Rogosa and Sharpe (MRS) medium to isolate Lactic acid bacteria, and a yeast extract peptone dextrose (YPD) (with kanamycin or ampicillin) medium to isolate yeast in soybean paste at various aging periods.

The *Doenjang* products (approx. 10 g each) were transferred to stomacher bags under aseptic conditions, mixed with 90 mL sterile saline solution and homogenized by stomacher blender for 1 min at top speed. Samples were centrifuged at 1500× *g* for 10 min at 4 °C. Supernatants were serially diluted in sterile saline solution and 100 µL of samples were spread on the following media: TSB agar for *Bacillus*, MRS agar for lactic acid bacteria, and YPD agar containing 25 µg/mL of ampicillin and 10 µg/mL of kanamycin for yeast. The agar plates were incubated at 37 °C under aerobic conditions for 16 h, and then isolated according to the shape and color of the colonies. 16S rRNA sequencing was performed to identify the isolated bacteria.

### 4.3. Identification of Isolated Strains

Bacterial genomic DNA samples were extracted using an InstaGenetm Matrix BIO-RAD (Hercules, CA, USA). To identify the phylogeny of the 16S rRNA from the strains, 27F (5′-AGAGTTTGATCCTGGCTCA G-3′) and 1492R (5′-GGTTACCTTGTTACGACTT-3′ [33] were used as universal primers to amplify fragments by polymerase chain reaction (PCR). The PCR reaction was performed with 20 ng of genomic DNA as a template in a 30 µL reaction mixture by using EF-Taq (SolGent, Daejeon, Korea) for the following cycles: activation of Taq polymerase at 95 °C for 2 min, followed by 35 cycles at 95 °C, 55 °C, and 72 °C for 1 min each, finishing with a 10 min step at 72 °C.

The amplified fragments were purified with a multiscreen filter plate (Millipore Corps., Billerica, MA, USA). A sequencing reaction was performed using a PRISM BigDye Terminator v3.1 Cycle sequencing kit (Applied Biosystems, Foster City, CA, USA) [33]. The DNA samples containing the extension products were added to Hi-Di formamide (Applied Biosystems, Foster City, CA, USA). The mixture was incubated at 95 °C for 5 min, followed by 5 min on ice, and then analyzed by an ABI Prism 3730XL DNA analyzer (Applied Biosystems, Foster City, CA, USA). Further, 16S rRNA sequencing was performed in Macrogen (Seoul, Korea).

Each 16S rRNA sequence of isolated bacteria was analyzed with the NCBI database using the BLASTN program [34] and the Seqmatch program (version 3) of the Ribosomal Database Project (RDP) of standard strains. For comparison, the CLUSTAL W program [35] was used.

### 4.4. Protease Assay

0.25 mL of cultured medium and 0.6% casein solution were dissolved in 0.5 M phosphate buffer (pH 7.0), then mixed and reacted at 37 °C for 30 min. Then, 0.5 mL of 0.44 M trichloroacetic acid was added to halt the reaction by leaving it to stand for 30 min at 4 °C. Subsequently, the reactant was centrifuged at 3000× *g* for 10 min, and the protein contents were assayed using the Folin method. The protease activity unit was defined as the amount of enzyme that produced 1 μg of tyrosine per min [36].

### 4.5. Amylase Assay

α-Amylase activity was assayed by the Dextrinogenic unit of the Nagase (DUN) method [37]. Then, 1 mL of cultured medium was added to 3 mL of 1% soluble starch (in a 0.02 M phosphate buffer, pH 7.0), and reacted at 40 °C for 10 min. Then, 10 mL of 1 M HCl was added to halt the reaction. After adding 10 mL of iodine solution (0.005% I_2_ + 0.05% KI), the absorbance was measured at 660 nm. The amylase activity unit was defined as the amount of enzyme that produced 1 mg of starch for 1 min.

### 4.6. Lipase Assay

A mixture of 1 mL of cultured medium, 1.0 mL of 0.5 M KCl, 1.0 mL of 5 mM CaCl_2,_ and 7.3 mL of distilled water was preincubated at 35 °C for 10 min, followed by reaction with 0.2 mL of tributyrin for 15 min. The reactant was stopped by boiling for 2 min, and the fatty acid produced was titrated with 0.01 N NaOH. The lipase activity unit was defined as the amount of enzyme that produced 1 μg of fatty acid for 1 min. [38].

### 4.7. Descriptive Analysis and Sensory Evaluation

A descriptive analysis of the aroma of the fermented products was conducted by twenty well-trained panels, and the descriptive attributes were derived.

### 4.8. Extraction of Volatiles by the Headspace Solid-Phase Microextraction (HS-SPME)

Modified methods of extracting volatile compounds from cultured or fermented products were used based on Lee and Ahn [18]. The sample (1 g) was placed in 30 mL vials (Supelco, Bellefonte, PA, USA) which were sealed with septum (silicon/Teflon, Supelco) after adding 3.5 mL of distilled water and 1 g of NaCl. The sealed vials were equilibrated by continuous stirring at 60 °C for 30 min, and then collected in a 50/30 μm DVB/carboxen™/PDMS StableFlex™ iber (Supelco) for 30 min and injected into the GC for 1 min.

### 4.9. Analysis of Volatiles by GC/MS

The volatile flavor compounds extracted by HS-SPME were analyzed by GC/MS (Agilent 6890 gas chromatography coupled with a 5973 mass selective detector, Agilent Co., Palo Alto, CA, USA) using a Stabilwax^®^-DA column (30 m length × 0.25 mm i.d. × 0.25 μm film thickness: Restek, Bellefonte, PA, USA). The oven temperature was kept constant at 40 °C for 3 min, after which it was increased to 210 °C at intervals of 4 °C/min. The injector temperature was held at 250 °C for the analysis of the split-less mode, and the flow rate of the carrier gas was held at 0.8 mL/min using helium. MSD (Agilent 5975C) conditions (capillary direct interface temperature 250 °C, ion source temperature 230 °C, EI ionization voltage 70 eV, mass range 40–550 atomic mass unit (amu) and scan rate 2.2 scans/s) were used, and the identification of the volatile compounds was confirmed by comparing the retention indices (RI), aromatic spectra (NIST05A), and aromatic properties.

### 4.10. The Ultra-High Performance Liquid Chromatography-Quadrupole Time-of-Flight Mass Spectrometry (UPLC-Q TOF-MS) Analysis

*Doenjang* powder was extracted with water (20 mg/400 µL) containing 8-Bromoguanosine (40 µg/mL) as the internal standard using a bullet blender (Next Advance, Troy, NY, USA). The samples were then homogenized with 50% acetonitrile (ACN) (25 mg/mL) and 70% methanol (15 mg/700 µL) with an internal standard (terfenadine) using a bullet blender. After centrifugation, the supernatants were analyzed using an UPLC-Q-TOF MS (Waters, Milford, MA, USA). A UPLC-Q-TOF (Waters, USA) was connected to an Acquity BEH C18 column (2.1 mm × 100 mm, 1.7 µm; Waters) at 40 °C. Deionized water containing 0.1% formic acid (A) and ACN containing 0.1% formic acid (B) at a flow rate of 0.35 mL/min was used as the mobile phase. The column eluents were detected with a Q-TOF MS with positive electrospray ionization (ESI). The de-solvation and source temperature were set to 400 °C and 120 °C, respectively; the de-solvation gas flow rate was set to 800 L/h; and the capillary and sampling cone voltages were set at 3 kV and 40 V, respectively.

Leucine-enkephaline ([M + H] = 556.2771) was used as a lock mass of the reference compound at a frequency of 10 s. MS data were obtained in the scan rage from 50 to 1500 *m/z* with a scan time of 0.2 s. The MS/MS data were obtained in the *m*/*z* 50–1500 using collision energy ramps from 20 to 40 eV. MassLynx software (Waters) was used for the data processing of the mass spectrometry data, including *m*/*z*, retention time, and ion intensity.

### 4.11. Statistical Analysis

Principle component analysis were performed to identify the correlation between the fermented samples and aroma compounds using the XLSTAT ver. 2007.1 (Addinsoft, New York, NY, USA). One-way ANOVA was performed to test for significant differences between each samples, and a Duncan’s multiple range test was performed to determine whether there were any significant differences between their mean values (*p* < 0.05).

## 5. Conclusions

This study aimed to investigate the characteristics of the microorganisms involved in the formation of the desirable flavor of *Doenjang*. Fourteen strains were isolated from traditional *Doenjang* products and a sensory evaluation was conducted on the aroma of each culture. Most of the strains showed relatively high enzyme activities, such as lipase, amylase and protease, and produced a desirable flavor, such as acetoin and phenethyl alcohol including more peptides. It is expected that these strains will be applied to the development of *Doenjang* products with more desirable flavor.

## Figures and Tables

**Figure 1 molecules-26-05067-f001:**
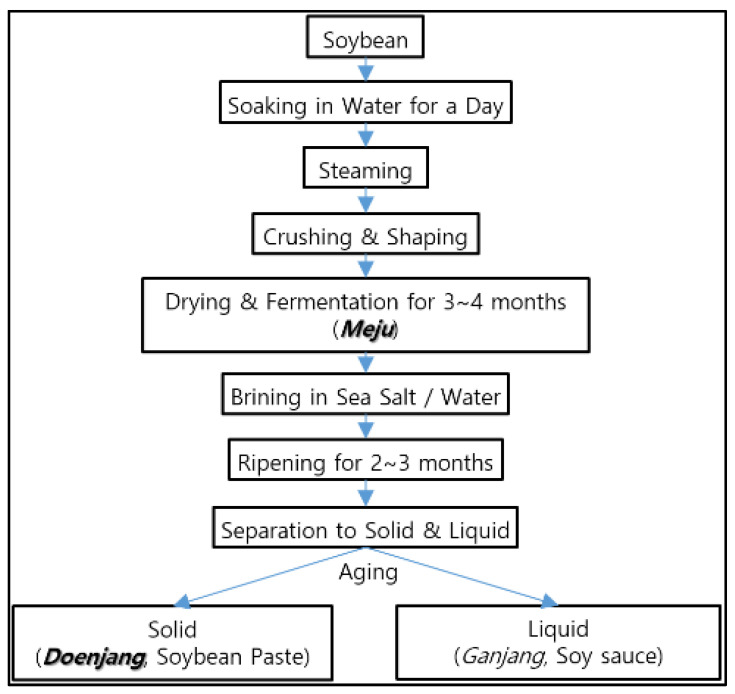
The general manufacturing flow of traditional *Doenjang*.

**Figure 2 molecules-26-05067-f002:**
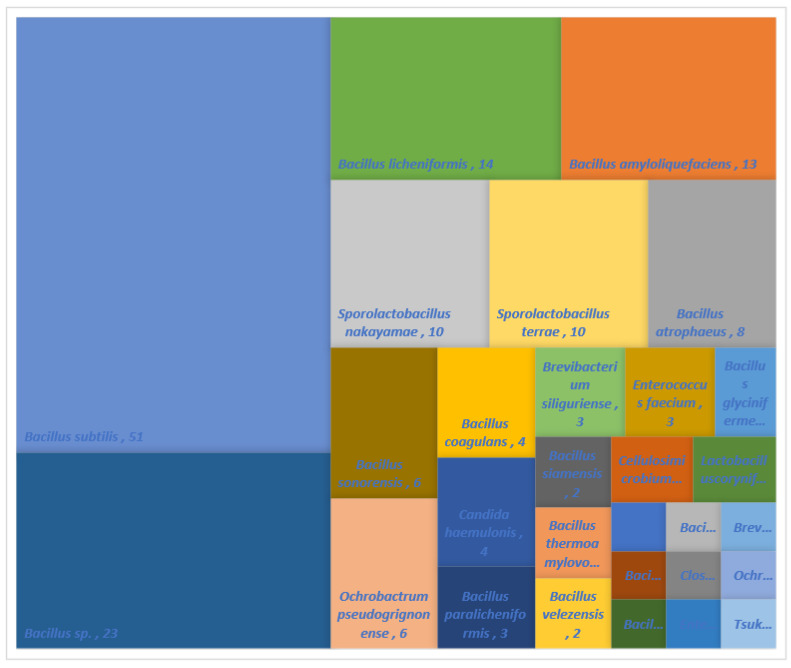
Ratio of screened strains from *Doenjang* based on 16S rRNA.

**Figure 3 molecules-26-05067-f003:**
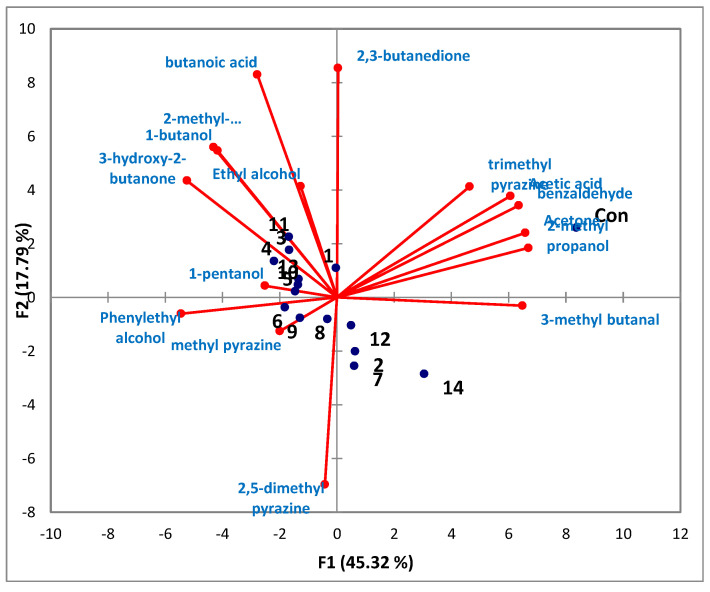
Principal component analysis of volatiles identified from each strain culture. 1. *B. sonorensis*, 2. *B. subtilis*-H1, 3. *B. subtilis*-H2, 4. *B. subtilis*-H3, 5. *B. subtilis*-H4, 6. *B. subtilis*-H5, 7. *B. subtilis*-H6, 8. *B. amyloliquefacience*-J1, 9. *B. amyloliquefacience*-J2, 10. *Bacillus*. sp-K1, 11. *Bacillus*. sp-K2, 12. *Bacillus*. sp-K3, 13. *Bacillus*. sp-K4, 14. *B. Frigoritolerance*.

**Figure 4 molecules-26-05067-f004:**
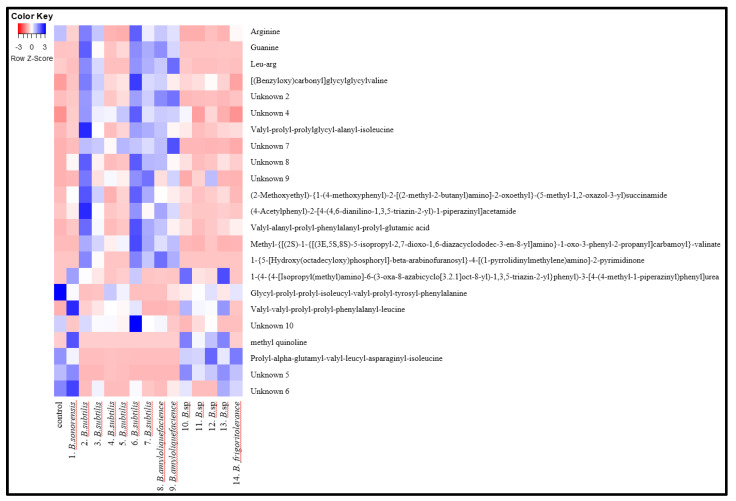
Heat map of non-volatile compounds of culture.

**Table 1 molecules-26-05067-t001:** Enzyme activities of various strains isolated from *Doenjang*.

Strains	Enzymes (Units)		
	Lipase	Amylase	Protease
*B. sonorensis*	982.2 ± 19.9 ^b^	0 ^a^	221.4 ± 5.1 ^a,b^
*B. subtilis*-H1	1090.9 ± 20.3 ^d^	171.8 ± 7.0 ^b^	272.6 ± 5.1 ^d,e^
*B. subtilis*-H2	1047.6 ± 50.1 ^c,d^	0 ^a^	276.6 ± 14.3 ^d,e,f^
*B. subtilis*-H3	988.6 ± 32.1 ^b,c^	0 ^a^	285.0 ± 6.7 ^e,f^
*B. subtilis*-H4	1015.6 ± 30.8 ^b,c^	0 ^a^	241.8 ± 12.2 ^c^
*B. subtilis*-H5	1023.6 ± 13.8 ^b,c^	0 ^a^	255.8 ± 6.8 ^c,d^
*B. subtilis*-H6	1355.2 ± 50.6 ^f^	4105.5 ± 105.4 ^f^	292.6 ± 7.5 ^e,f,g^
*B. amyloliquefacience*-J1	1251.8 ± 8.6 ^e^	0 ^a^	279.1 ± 2.9 ^d,e,f^
*B. amyloliquefacience*-J2	1310.3 ± 19.8 ^f^	242.6 ± 8.4 ^c^	319.5 ± 2.9 ^h^
*B*. sp-K1	2453.6 ± 51.2 ^i^	0 ^a^	206.0 ± 8.8 ^a^
*B*. sp-K2	1639.9 ± 39.8 ^h^	0 ^a^	300.9 ± 13.7 ^f,g,h^
*B*. sp-K3	1558.7 ± 12.2 ^g^	860.0 ± 20.9 ^d^	317.1 ± 8.2 ^g,h^
*B*. sp-K4	1588.9 ± 9.8 ^g,h^	271.0 ± 9.4 ^c^	317.3 ± 44.1 ^g,h^
*B. frigoritolerance*	924.9 ± 41.6 ^a^	0 ^a^	221.2 ± 4.5 ^a,b^

^a–h^ Means on the same column with different letters are significantly different (*p* < 0.05).

**Table 2 molecules-26-05067-t002:** Volatile composition of cultures by GC/MS based on area (%).

Compounds (Odor)	Control	*B. sonorensis*	*B.subtilis*-H2	*B.subtilis*-H3	*B.subtilis*-H4	*B.subtilis*-H5	*B.subtilis*-H6	*B.subtilis*-H7	*B.amyloliquefaciens*-J2	*B.amyloliquenfaciens*-J3	*B.*sp-K2	*B.*sp-K3	*B.*sp-K4	*B.*sp-K5	*B. frigoritolerance*
Ethyl alcohol	0.14	3.17	0.00	0.00	0.00	0.00	0.00	0.00	0.00	0.00	2.47	1.88	0.66	2.15	0.02
Acetone	17.23	7.71	10.27	4.47	4.89	5.36	5.24	9.12	8.15	5.83	4.94	5.60	8.67	4.84	9.92
2-Methyl propanol	5.07	1.52	1.30	0.35	0.47	0.48	0.56	2.37	1.00	0.50	0.90	0.94	1.75	0.83	2.99
2,3-Butanedione (Buttery, cheese)	7.44	8.65	9.70	8.69	8.93	8.43	7.71	9.41	10.03	8.45	7.70	9.18	7.16	7.62	7.20
3-Methyl butanal (Malty)	26.39	14.79	16.31	6.96	6.09	8.41	8.03	15.90	13.77	8.50	11.94	12.62	21.19	10.92	32.00
Acetic acid (Sour)	11.27	9.06	6.82	5.50	5.16	5.80	5.43	6.60	6.02	4.63	6.41	6.85	9.27	6.01	10.62
1-Butanol (Soap, Fatty, diesel)	0.94	15.44	2.23	8.79	11.42	11.01	11.43	2.15	3.41	6.99	12.59	13.93	15.53	11.63	1.43
3-Hydroxy-2-butanone (Acetoin, Buttery)	1.62	14.91	23.51	43.37	41.01	33.69	35.05	17.63	30.04	37.53	25.47	35.23	3.63	30.50	2.27
1-Pentanol	0.42	1.61	0.98	0.99	1.03	0.65	1.27	1.16	1.49	0.82	0.75	0.78	1.04	0.71	0.85
2-Methyl-propanoic acid	0.41	0.65	0.59	1.24	1.06	1.21	0.71	0.98	0.76	0.88	1.09	1.30	0.99	1.09	0.70
Methyl pyrazine (Nutty, roasted)	2.60	3.48	4.19	2.98	3.18	3.66	3.60	4.72	3.89	3.85	3.62	3.78	3.93	3.41	4.02
Butanoic acid (Sour)	0.45	0.88	0.51	0.54	0.69	0.75	0.74	0.25	0.59	0.61	0.68	0.72	0.67	0.68	0.18
2,5-Dimethyl pyrazine (Nutty, roasted)	8.16	14.26	18.59	11.81	12.61	16.29	15.77	22.58	16.51	16.73	16.64	2.16	18.62	15.27	19.21
Benzaldehyde (Sweet, candy, wood)	15.45	1.83	2.11	1.82	1.32	1.70	1.78	3.61	1.73	1.86	2.08	2.15	4.06	1.88	5.94
Trimethyl pyrazine (Nutty, roasted)	2.40	1.91	2.46	2.13	1.77	2.19	2.15	3.01	2.22	2.36	2.25	2.26	2.37	2.09	2.40
Phenylethyl alcohol (Rose)	0.02	0.12	0.43	0.36	0.37	0.39	0.55	0.52	0.37	0.47	0.48	0.63	0.47	0.39	0.26
Total	100.00	100.00	100.00	100.00	100.00	100.00	100.00	100.00	100.00	100.00	100.00	100.00	100.00	100.00	100.00

**Table 3 molecules-26-05067-t003:** Identification of major metabolites contributing to separation among the sample groups.

No.	RT (min)	Compound	Exact Mass	MS Fragments	VIP	*p*-Value
			(M + H)			
1	0.69	Arginine	175.1190		1.26	1.66 × 10^−4^
2	1.01	Guanine	152.0572	135	1.56	2.82 × 10^−5^
3	1.43	Leu-arg	288.2019	158	1.47	1.82 × 10^−5^
4	1.70	Adenine	136.0762		1.20	5.73 × 10^−2^
5	2.61	Unknown 1	343.1979		0.99	1.43 × 10^−1^
6	2.89	[(Benzyloxy)carbonyl]glycylglycylvaline	366.1685	203, 130	1.25	2.49 × 10^−2^
7	3.02	Unknown 2	515.2479		1.44	2.38 × 10^−4^
8	3.05	Unknown 3	344.1820		1.19	7.59 × 10^−2^
9	3.12	Unknown 4	442.2668		1.29	2.46 × 10^−3^
10	3.51	Methyl quinoline	144.0820		1.53	5.28 × 10^−5^
11	3.60	Unknown 5	508.2507		1.53	4.56 × 10^−6^
12	3.81	Unknown 6	288.1925		1.18	8.14 × 10^−7^
13	4.00	Valyl-prolyl-prolylglycyl-alanyl-isoleucine	553.3373	147, 244	1.34	2.05 × 10^−3^
14	4.02	Unknown 7	585.3271		1.25	1.41 × 10^−4^
15	4.23	Unknown 8	560.2731		1.50	1.67 × 10^−4^
16	4.27	Unknown 9	360.1964		1.03	1.10 × 10^−2^
17	4.43	(2-Methoxyethyl)-{1-(4-methoxyphenyl)-2-[(2-methyl-2-butanyl)amino]-2-oxoethyl}-(5-methyl-1,2-oxazol-3-yl)succinamide	489.2714	374, 279	1.39	1.17 × 10^−3^
18	4.71	Prolyl-alpha-glutamyl-valyl-leucyl-asparaginyl-isoleucine	684.3945	427, 555	1.28	1.60 × 10^−10^
19	5.04	(4-Acetylphenyl)-2-[4-(4,6-dianilino-1,3,5-triazin-2-yl)-1-piperazinyl]acetamide	523.2538	360	1.40	4.76 × 10^−4^
20	5.16	Valyl-alanyl-prolyl-phenylalanyl-prolyl-glutamic acid	659.3424	70, 227	1.57	7.73 × 10^−7^
21	5.21	Methyl-{[(2S)-1-{[(3E,5S,8S)-5-isopropyl-2,7-dioxo-1,6-diazacyclododec-3-en-8-yl]amino}-1-oxo-3-phenyl-2-propanyl]carbamoyl}-valinate	558.3306	263, 360	1.42	5.27 × 10^−4^
22	5.69	1-{5-[Hydroxy(octadecyloxy)phosphoryl]-beta-arabinofuranosyl}-4-[(1-pyrrolidinylmethylene)amino]-2-pyrimidinone	657.4005	583	1.59	2.67 × 10^−7^
23	5.82	1-(4-{4-[Isopropyl(methyl)amino]-6-(3-oxa-8-azabicyclo [3.2.1]oct-8-yl)-1,3,5-triazin-2-yl}phenyl)-3-[4-(4-methyl-1-piperazinyl)phenyl]urea	572.3463	554	1.44	8.01 × 10^−4^
24	6.08	Glycyl-prolyl-prolyl-isoleucyl-valyl-prolyl-tyrosyl-phenylalanine	889.4830	507, 627	1.13	7.75 × 10^−10^
25	6.09	Valyl-valyl-prolyl-prolyl-phenylalanyl-leucine	671.4117	217, 376, 473	1.57	2.19 × 10^−5^
26	6.76	Unknown 10	274.2727		1.07	3.68 × 10^−8^

## Data Availability

The data presented in this study are available on request from the corresponding author.

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
