# Peer review of "Characteristics of Bacterial Strains with Desirable Flavor Compounds from Korean Traditional Fermented Soybean Paste (Doenjang)"

_molecules, 2021, doi:10.3390/molecules26165067_

Round 1
Reviewer 1 Report
1.Introduction/Discussion: The proper production of Doenjang presumably requires a mixture of various microorganisms and the authors should therefore explain better the goal of the paper and how the results could be utilized in the food industry.
2.The authors should check and correct the use of italics (meju, microorganisms) and capitals (e.g. compounds in Table 2: 2-Methyl… etc.).
3.Many abbreviations are not explained (GABA, PCA, HS-SPME, GC, MS, MSD, CAN, MRS, YPD, etc.).
4.Results and Methods. It is often unclear how the isolated strains were cultured and which material was analyzed for the metabolites. Did the authors analyzed some commercial media or media with soybean paste? The preparation of samples in Methods and Tables should be described better.
5.Description in Fig. 1 needs to be improved.
6.The names of peptides should use 3-letter abbreviations only.
7.In the whole text, the authors should avoid the use of vague terms such as ester (page 8, paragraph 5), sample, etc. – it must be specified.
- Methods are also often unclear. For example, how were the enzyme units calculated? How was trypsin quantified? Amylase cleaves, not produces starch.
Author Response
1.Introduction/Discussion: The proper production of Doenjang presumably requires a mixture of various microorganisms and the authors should therefore explain better the goal of the paper and how the results could be utilized in the food industry.
- Revised according to the comment of the reviewer
- Using this result, selected microorganisms could be applied in the production of soybean products with excellent butter, rose aroma and umami
2.The authors should check and correct the use of italics (meju, microorganisms) and capitals (e.g. compounds in Table 2: 2-Methyl… etc.).
- Revised according to the comment of the reviewer
3.Many abbreviations are not explained (GABA, PCA, HS-SPME, GC, MS, MSD, CAN, MRS, YPD, etc.).
- Revised according to the comment of the reviewer
4.Results and Methods. It is often unclear how the isolated strains were cultured and which material was analyzed for the metabolites. Did the authors analyzed some commercial media or media with soybean paste? The preparation of samples in Methods and Tables should be described better.
- Revised according to the comment of the reviewer
5.Description in Fig. 1 needs to be improved.
- Modified Figure 1
6.The names of peptides should use 3-letter abbreviations only.
- Revised according to the comment of the reviewer
7.In the whole text, the authors should avoid the use of vague terms such as ester (page 8, paragraph 5), sample, etc. – it must be specified.
- Although there are many types of esters, it is not possible to specify them, but the most detected ester in traditional doenjang is ethylacetate, ethylbutyrate, 2-methyl butanoate, etc. If there are many types of ester components and produced in large quantities, it can be said that the flavor is excellent
corrected “ester” to “ester like ethylacetate”
Methods are also often unclear. For example, how were the enzyme units calculated? How was trypsin quantified? Amylase cleaves, not produces starch.
- Revised according to the comment of the reviewer
Reviewer 2 Report
Though both the title and the abstract caught my interest, the results as presented does not been sufficient to support the claims and the discussion was in my opinion rather generic. The sensory evaluation is hardly discussed. Moreover, it is not clear at what point the culturing were the samples collected for the analysis of the volatile compounds and also why, how and to what extent there was a change in the distribution of these products in the course of the cultivations. It is noted in the introduction, which is well written and structured, indeed, that the aromas and flavours of the doenjang change during aging. That aspect is not discussed at all in the mains study. Ιn my opinion the final claim that the “these strains will be applied to the development of doenjang products with a desirable flavour” is rather a conjecture that is not supported by the analysis performed. I would suggest that the authors should re-evaluate their results on the basis of certain metabolic routes of the implicated species as well as the intraspecies interactions and make more in depth analysis of their data in order to support their claim.
Specific comments
Lines 78-82: You refer to bacterial and yeasts species i.e. Bacillus subtilis, Bacillus amyloliquefaciens, Bacillus licheniformis, etc and you mention that several species of those were identified. What do mean? Do you refer to strains?
Figue 1: Poor quality of figure. Please chain format or add legend so as the species names to be readable
Table 1: Where do you attribute the non amylolytic activity of B. amyloliquefacience-J1 ?
Lines 98-100: The argument is very generic. Ptroteases, lipases and amylases break down the respective polymers but such activity does not warranty the subsequent biotransormation of the monomers to aromatic compounds.
Lines 103-104: maybe species instead of types?
Lines 103-105 and table 2: The table shows the relative presence of certain componds at a certain point of culturing (??) of selective strains in monocultures. To what degree could those be assoiated to thw the final aromas/flavours of doenjang. I mean in a microbial consortium the interactions among species wouldn’t affect the final reults?
Lines 108-109: The meaning of the sentence is not totally clear. What do you mean by “lexocones”? Do you mean that the organolepric characteristics (odour?) of the cultures was verbally described on basis of specific foods?
Table 2:
It would be helpful if you provided the odour that each volatile compound is responsible for. Either in the table or in the text. The legend of the table should also be more informative. Pls revise.
Line 115-118: At what point of bacterial growth were the identified volatile compounds identified and when did they decrease? Where they secondarily bioconvered to other molecules that you mention below?
What do you mean by “after strain culture”. What is “a relatively large amount of …”? Have you quantified those compounds?
“…were generated in most of the strain cultures.” This is very generic
Line 119 “…the above results…” Do you refer to the findings presented in table 2?
Figure 2: The results presented in the figure are hardly discussed. What does this PCA analysis prove with respect to the bacterial strains and dominance of certain odours/flavours?
Lines 133-135: Please provide references to support your argument
Line 190: This is not correct
Lines 216-218: Very generic
Lines 225-228 and 336-338: How is hypothesis supported by the findings of the current study? Please analyse
Author Response
Though both the title and the abstract caught my interest, the results as presented does not been sufficient to support the claims and the discussion was in my opinion rather generic. The sensory evaluation is hardly discussed. Moreover, it is not clear at what point the culturing were the samples collected for the analysis of the volatile compounds and also why, how and to what extent there was a change in the distribution of these products in the course of the cultivations. It is noted in the introduction, which is well written and structured, indeed, that the aromas and flavours of the doenjang change during aging. That aspect is not discussed at all in the mains study. Ιn my opinion the final claim that the “these strains will be applied to the development of doenjang products with a desirable flavour” is rather a conjecture that is not supported by the analysis performed. I would suggest that the authors should re-evaluate their results on the basis of certain metabolic routes of the implicated species as well as the intraspecies interactions and make more in depth analysis of their data in order to support their claim.
- Since what we were dealing with was a pure strain culture medium, we could not perform a sensory evaluation.
- Strain cultures were not eligible for IRB approval
- In this study, the research team selected and studied strains with excellent flavor just by smelling the fermented product.
<Specific comments>
Lines 78-82: You refer to bacterial and yeasts species i.e. Bacillus subtilis, Bacillus amyloliquefaciens, Bacillus licheniformis, etc and you mention that several species of those were identified. What do mean? Do you refer to strains?
- We are speaking about the distribution of various microorganisms that show the characteristics of soybean paste. And among these, we want to apply microorganisms with excellent fermentative properties to the manufacture of other fermented soybean foods.
Figue 1: Poor quality of figure. Please chain format or add legend so as the species names to be readable
- Modified Figure 1
Table 1: Where do you attribute the non amylolytic activity of B. amyloliquefacience-J1 ?
- Although the activity was low under certain conditions such as this result, it can be applied to other fields if other fermentation properties such as protease and lipase avtivities are excellent.
Lines 98-100: The argument is very generic. Ptroteases, lipases and amylases break down the respective polymers but such activity does not warranty the subsequent biotransormation of the monomers to aromatic compounds.
- It is reported that proteases and lipases can produce fatty acids, which are precursors of free amino acids, organic acids, and volatiles, mainly during doenjang fermentation (kim and others 1992).
(Kim, G. E.; Kim, M. H.; Choi, B. D.; Kim, T. S.; Lee, J. H., Flavor compounds of domestic meju and doenjang(soybean paste). Food Chemistry 1992, 83, 339-342)
Lines 103-104: maybe species instead of types?
- Revised according to the comment of the reviewer
Lines 103-105 and table 2: The table shows the relative presence of certain componds at a certain point of culturing (??) of selective strains in monocultures. To what degree could those be assoiated to thw the final aromas/flavours of doenjang. I mean in a microbial consortium the interactions among species wouldn’t affect the final reults?
- The amount and composition of flavor components produced during the maturation process of doenjang are affected by various microorganisms. This study was to confirm the ability of each microorganism to produce aroma components and to examine the applicability to soybean paste.
Lines 108-109: The meaning of the sentence is not totally clear. What do you mean by “lexocones”? Do you mean that the organolepric characteristics (odour?) of the cultures was verbally described on basis of specific foods?
- Corrected “Lexicon” to “sensory characteristics”
Table 2:
It would be helpful if you provided the odour that each volatile compound is responsible for. Either in the table or in the text. The legend of the table should also be more informative. Pls revise.
- Revised according to the comment of the reviewer
Line 115-118: At what point of bacterial growth were the identified volatile compounds identified and when did they decrease? Where they secondarily bioconvered to other molecules that you mention below?
- The microorganism used in the experiment was a microorganism that entered the stationary phase, and the amount of the compound mentioned in the text was compared with the TSB medium.
What do you mean by “after strain culture”. What is “a relatively large amount of …”? Have you quantified those compounds?
- It is a comparison between the medium itself and the medium after culturing microorganisms.
“…were generated in most of the strain cultures.” This is very generic
- Corrected “generated” to “found“
Line 119 “…the above results…” Do you refer to the findings presented in table 2?
- Revised clear to understand
Figure 2: The results presented in the figure are hardly discussed. What does this PCA analysis prove with respect to the bacterial strains and dominance of certain odours/flavours?
- Distance between compounds and bacteria means their correlation and we focused on 3 major compound(3-hydroxy-2-butanone (acetoin) with butter or cheese notes, 2, 5-dimethyl pyrazine with nutty notes, phenethyl alcohol with rose notes)
Lines 133-135: Please provide references to support your argument
- Indicated in the text
Line 190: This is not correct
- Deleted
Lines 216-218: Very generic
- Generic, but helpful in understanding the overall content.
Lines 225-228 and 336-338: How is hypothesis supported by the findings of the current study? Please analyse
- Many findings concerning the volatile compounds related to Doenjang have been well documented [22, 31]. Relatively high concentrations of ethyl alcohol, isoamyl alcohol and phenethyl alcohol, including benzaldehyde, 3-methyl butanal, and ethyl acetate, as well as pyrazine are shown in the certified traditional Doenjang products [22].
- In this study, 2, 5-dimethyl pyrazine(nutty note) and phenethyl alcohol(rosy note) and 3-hydroxy-2-butanone(acetoin, buttery note) were mainly produced and they are one of the volatile compound in Doenjang. [31]. Also as they have preferable notes such as nutty, rosy and buttey note to doenjang, we expect that above compound is helpful to give desirable flavors to doenjang if the strain is applied to doenjang.
Reviewer 3 Report
Paper:
Characteristics of Bacterial Strains with Desirable Flavor Compounds from Korean Traditional Fermented Soybean Paste (Doenjang)
Journal: Molecules.
General comments:
In the present work a total of 134 strains to the species Bacillus spp. were isolated from doenjang, a Korean traditional fermented soybean paste. Fourteen of 134 strains isolated showed enzymatic activity (lipase, amylase and protease). These strains were tested in doenjang production and the final product was evaluated for sensory attributes an volatile organic compounds. Although I am personally involved in the valorisation of traditional foods and the topic falls within the topics on Molecules Journal, the paper showed several weaknesses and n my opinion, is not ready to be published in the present form.
Specific comment:
Title: In my opinion, the Title should report a statement, a conclusion, in the present form seems a “main hypothesis” around which the work is developed.
Abstract: The abstract needs to be reordered and it does not stand alone the way it is actually written. There is need to read the complete manuscript to understand the abstract. However, in general the abstract should be more descriptive, reporting the main numerical data (percentages, levels, concentrations etc.).
Introduction: (line 36). A flow diagram depicting the manufacturing procedure of this product should help to understand its technology, and would allow comparison with the manufacturing process of other similar products.
Main hypothesis: (lines 65-67). Should be rewritten in order to better clarify (if I understand) that the object of the work was to isolate autochthonous Bacillus spp. strains from doenjang and than use the strains that showed the best performance as starter culture in order to improve the aromatic characteristic of this fermented traditional product.
M&M: (lines 245-249). Regarding the isolation plan number of samples were provided while are not reported information about of the repetition and the characteristics of the samples chosen.
(line 260). Information about the plate count and incubation conditions for Bacillus spp. were provided while apart from the grow media these information are absent for the lactic acid bacteria and yeasts.
(line 293). If I understand the judge evaluated the doenjang products produced with the single inocula of the 14 Bacillus that showed the pest performance in term of enzymatic activity. Furthermore, Is this panel appropriate? Were thy trained? Please provide details. However, I would like to ask the authors if during sensory evaluation the judges were able to see the samples of there was a blind test.
Results: (paragraph 2.1). The amplification and sequencing of the 16S rRNA gene does not confirm that the isolates identified are different strains you should be applied an strain typing by randomly amplified polymorphic DNA (RAPD)-PCR technique. Also include that sequence data were deposited in Genbank.
(Fig. 1). It is confused. You could replace it with a table.
There are some other minor points, but I think that there is no point in mentioning them at this stage.
Author Response
In the present work a total of 134 strains to the species Bacillus spp. were isolated from doenjang, a Korean traditional fermented soybean paste. Fourteen of 134 strains isolated showed enzymatic activity (lipase, amylase and protease). These strains were tested in doenjang production and the final product was evaluated for sensory attributes an volatile organic compounds. Although I am personally involved in the valorisation of traditional foods and the topic falls within the topics on Molecules Journal, the paper showed several weaknesses and n my opinion, is not ready to be published in the present form.
- Here, it was not an experiment applied to soybean paste(doenjang), but an experiment conducted with a microbial culture. Afterwards, we will apply the results of this experiment to soybean paste(doenjang),.
<Specific comment:>
Title: In my opinion, the Title should report a statement, a conclusion, in the present form seems a “main hypothesis” around which the work is developed.
- We hope to maintain title as it is
Abstract: The abstract needs to be reordered and it does not stand alone the way it is actually written. There is need to read the complete manuscript to understand the abstract. However, in general the abstract should be more descriptive, reporting the main numerical data (percentages, levels, concentrations etc.).
- Revised according to the comment of the reviewers
Introduction: (line 36). A flow diagram depicting the manufacturing procedure of this product should help to understand its technology, and would allow comparison with the manufacturing process of other similar products.
- Revised according to the comment of the reviewer
Main hypothesis: (lines 65-67). Should be rewritten in order to better clarify (if I understand) that the object of the work was to isolate autochthonous Bacillus spp. strains from doenjang and than use the strains that showed the best performance as starter culture in order to improve the aromatic characteristic of this fermented traditional product.
- Revised according to the comment of the reviewer
M&M: (lines 245-249). Regarding the isolation plan number of samples were provided while are not reported information about of the repetition and the characteristics of the samples chosen.
- Revised according to the comment of the reviewer
(line 260). Information about the plate count and incubation conditions for Bacillus spp. were provided while apart from the grow media these information are absent for the lactic acid bacteria and yeasts.
- Revised according to the comment of the reviewer
(line 293). If I understand the judge evaluated the doenjang products produced with the single inocula of the 14 Bacillus that showed the pest performance in term of enzymatic activity. Furthermore, Is this panel appropriate? Were thy trained? Please provide details. However, I would like to ask the authors if during sensory evaluation the judges were able to see the samples of there was a blind test.
- Since what we were dealing with was a pure strain culture medium, we could not perform a sensory evaluation.
- Strain cultures were not eligible for IRB approval
- In this study, the research team selected and studied strains with excellent flavor just by smelling the fermented product.
Results: (paragraph 2.1). The amplification and sequencing of the 16S rRNA gene does not confirm that the isolates identified are different strains you should be applied an strain typing by randomly amplified polymorphic DNA (RAPD)-PCR technique. Also include that sequence data were deposited in Genbank.
- Revised according to the comment of the reviewer
(Fig. 1). It is confused. You could replace it with a table.
- Modified Figure 1
There are some other minor points, but I think that there is no point in mentioning them at this stage.
-We tried to do our best to make it better
Reviewer 4 Report
This paper provides some interesting information on the identification and screening of strains (mainly bacteria) capable of producing aromatic compounds of interest in a traditional product, doenjang. Despite this, it was not clear how the strains studied in culture medium would maintain their enzymatic activity in the finished product and/or during storage. In general, microorganisms are able to modulate their metabolism, as well as their enzyme production performance, as a function of environmental conditions, i.e. nutrients, temperature, pH, water activity... Thus, it is essential to carry out the study under formulation conditions of the food product, or as closely as possible. Still, the paper proposes the study of descriptive and sensorial analysis of the product in section 4.7 of the M&M, but they were not presented. This data could contribute to evaluate the real feasibility of using these microorganisms to improve the flavor of doenjang, and would bring more robustness to the study. I suggest authors check these points before the paper could be considered for publication.
Line 39: Genus not 'species', also genus names should be italicized. Check the whole manuscript for the correct scientific nomenclature.
Line 78-82: This paragraph needs to be corrected. For example, "Thirty-one isolates belonging to the species Bacillus subtilis have been identified." Check the whole manuscript.
Lines 86-87: What authors means by 'good scent'? Please rewrite.
Line 88: Higher protease activity compared to what?
Line 95: High, not 'higher'.
Lines 153-157: Strains with high enzymatic activity were selected in culture medium, not in doenjang. How can the authors ensure that these strains will maintain the same enzymatic activity, i.e. good production of certain flavors of interest, in the final product? Please clarify.
Line 208: 'strain cultures found in this study'
Lines 239-241: This sentence needs to be reformulated.
Lines 254-255: Please, specify the sample/supernatant volume or mass.
Line 264: Please, specify details of PCR conditions.
Line 270: Authors deposited the 16S rRNA gene sequences in the NCBI-GenBank? Please provide them.
Lines 283-285: Delete, repeated information.
Line 234: The results of the sensory analysis were not presented, so this statement is speculative and should be deleted.
Lines 346-347: Again, the data presented were conducted in culture medium and strains may not represent the expected activity in the final product (doenjang).
Figure 1 needs to be modified, the caption should be placed below to avoid overlapping text.
Author Response
This paper provides some interesting information on the identification and screening of strains (mainly bacteria) capable of producing aromatic compounds of interest in a traditional product, doenjang. Despite this, it was not clear how the strains studied in culture medium would maintain their enzymatic activity in the finished product and/or during storage. In general, microorganisms are able to modulate their metabolism, as well as their enzyme production performance, as a function of environmental conditions, i.e. nutrients, temperature, pH, water activity... Thus, it is essential to carry out the study under formulation conditions of the food product, or as closely as possible. Still, the paper proposes the study of descriptive and sensorial analysis of the product in section 4.7 of the M&M, but they were not presented. This data could contribute to evaluate the real feasibility of using these microorganisms to improve the flavor of doenjang, and would bring more robustness to the study. I suggest authors check these points before the paper could be considered for publication.
- This study is a research on microorganisms to make fermented soybean products. Therefore, we have isolated and analyzed the characteristics of microorganisms in fermented soybean food, which are excellent in sensory quality.
- Strain cultures were not eligible for IRB approval
- In this study, the research team selected and studied strains with excellent flavor just by smelling the fermented product.
Line 39: Genus not 'species', also genus names should be italicized. Check the whole manuscript for the correct scientific nomenclature.
- Revised according to the comment of the reviewer
Line 78-82: This paragraph needs to be corrected. For example, "Thirty-one isolates belonging to the species Bacillus subtilis have been identified." Check the whole manuscript.
- Revised according to the comment of the reviewer
Lines 86-87: What authors means by 'good scent'? Please rewrite.
- As stated in the text, it is based on "rose", "butter", "nutty" aroma
Line 88: Higher protease activity compared to what?
- Revised according to the comment of the reviewer
Line 95: High, not 'higher'.
- Revised according to the comment of the reviewer
Lines 153-157: Strains with high enzymatic activity were selected in culture medium, not in doenjang. How can the authors ensure that these strains will maintain the same enzymatic activity, i.e. good production of certain flavors of interest, in the final product? Please clarify.
- The main component of the TSB medium used for bacterial isolation was soybean, and it is expected that the enzyme activity will appear in soybean fermented foods using the selected strain. Not all possibilities could be investigated at the strain screening stage.
Line 208: 'strain cultures found in this study'
- Revised according to the comment of the reviewer
Lines 239-241: This sentence needs to be reformulated.
- Revised according to the comment of the reviewer
Lines 254-255: Please, specify the sample/supernatant volume or mass.
- Revised according to the comment of the reviewer
Line 264: Please, specify details of PCR conditions.
- Revised according to the comment of the reviewer
Line 270: Authors deposited the 16S rRNA gene sequences in the NCBI-GenBank? Please provide them.
- Revised according to the comment of the reviewer
Lines 283-285: Delete, repeated information.
- Revised according to the comment of the reviewer
Line 234: The results of the sensory analysis were not presented, so this statement is speculative and should be deleted.
- Strain cultures were not eligible for IRB approval
Lines 346-347: Again, the data presented were conducted in culture medium and strains may not represent the expected activity in the final product (doenjang).
- As, the main ingredient of the TSB medium used for bacterial isolation was soybean, and it is expected that the enzyme activity will appear in soybean fermented foods(Doenjang) using the selected strain. Not all possibilities could be investigated at the strain screening stage.
Figure 1 needs to be modified, the caption should be placed below to avoid overlapping text.
- Modified Figure 1
Round 2
Reviewer 1 Report
Check the abbreviations and compound names once more.
TSB first appears on line 76, but it is explained on line 106. Do not use capital letters in retention time (line 151). GABA is expleined repeatedly and gamma should be written with a Greek letter (lines 64,196). Correct Doenjang on lines 263-264.Author Response
- Revised according to your kind comments
Reviewer 3 Report
The authors improved the quality of the manuscript but did not answer all my questions.
e.g. I asked for a more descriptive abstract reporting the main numerical data (percentages, levels, concentrations etc.) and it was not done.
However, the main criticisms of the work remain given by the fact that the authors speak of strains without having carried out a molecular strain typing. Furthermore, they have not entered the accession numbers of deposit on Genebank.
Author Response
- Revised according to your kind comments
- In response to the second comment, 16s rRNA sequencing was performed to identify strains isolated from Doenjang, and the sequences were analyzed using the Genebank database as reference sequence. There is no Genebank accession number because nucleotide sequences other than the 16s rRNA gene for identification were not analyzed. The molecular strain typing you requested is what we will do in the future for commercialization.
Reviewer 4 Report
The authors adequately addressed my questions raised during the review of the paper. Thus, I believe the paper can now be considered for publication.
Author Response
Thanks for your comments to get this work done